# Fruit Detection and Yield Mass Estimation from a UAV Based RGB Dense Cloud for an Apple Orchard

**Marius Hobart** [1,*], **Michael Pflanz** [2], **Nikos Tsoulias** [3], **Cornelia Weltzien** [1,4], **Mia Kopetzky** [5] and **Michael Schirrmann** [1]

1   Leibniz Institute for Agricultural Engineering and Bioeconomy, Max-Eyth-Allee 100, 14469 Potsdam, Germany; cweltzien@atb-potsdam.de (C.W.); mschirrmann@atb-potsdam.de (M.S.)
2   Landesamt für Umwelt, Seeburger Chaussee 2, 14476 Potsdam, Germany; michael.pflanz@lfu.brandenburg.de
3   Institut für Technik, Geisenheim University, Von-Lade-Straße 1, 65366 Geisenheim, Germany; nikolaos.tsoulias@hs-gm.de
4   FG Agromechatronik, Institut für Maschinenkonstruktion und Systemtechnik, Technische Universität Berlin, Straße des 17. Juni 144, 10623 Berlin, Germany
5   School of Sustainability, Leuphana-Universität Lüneburg, Universitätsallee 1, 21335 Lüneburg, Germany; mia.kopetzky@stud.leuphana.de
*   Correspondence: mhobart@atb-potsdam.de; Tel.: +49-331-5699-444

**Abstract:** Precise photogrammetric mapping of preharvest conditions in an apple orchard can help determine the exact position and volume of single apple fruits. This can help estimate upcoming yields and prevent losses through spatially precise cultivation measures. These parameters also are the basis for effective storage management decisions, post-harvest. These spatial orchard characteristics can be determined by low-cost drone technology with a consumer grade red-green-blue (RGB) sensor. Flights were conducted in a specified setting to enhance the signal-to-noise ratio of the orchard imagery. Two different altitudes of 7.5 m and 10 m were tested to estimate the optimum performance. A multi-seasonal field campaign was conducted on an apple orchard in Brandenburg, Germany. The test site consisted of an area of 0.5 ha with 1334 trees, including the varieties 'Gala' and 'Jonaprince'. Four rows of trees were tested each season, consisting of 14 blocks with eight trees each. Ripe apples were detected by their color and structure from a photogrammetrically created three-dimensional point cloud with an automatic algorithm. The detection included the position, number, volume and mass of apples for all blocks over the orchard. Results show that the identification of ripe apple fruit is possible in RGB point clouds. Model coefficients of determination ranged from 0.41 for data captured at an altitude of 7.5 m for 2018 to 0.40 and 0.53 for data from a 10 m altitude, for 2018 and 2020, respectively. Model performance was weaker for the last captured tree rows because data coverage was lower. The model underestimated the number of apples per block, which is reasonable, as leaves cover some of the fruits. However, a good relationship to the yield mass per block was found when the estimated apple volume per block was combined with a mean apple density per variety. Overall, coefficients of determination of 0.56 (for the 7.5 m altitude flight) and 0.76 (for the 10 m flights) were achieved. Therefore, we conclude that mapping at an altitude of 10 m performs better than 7.5 m, in the context of low-altitude UAV flights for the estimation of ripe apple parameters directly from 3D RGB dense point clouds.

**Keywords:** fruit detection; yield estimation; structure from motion (SfM); unmanned aerial vehicle (UAV); apple trees

## 1. Introduction

In commercial fruit orchards, timely information can help optimize management decisions, potentially increasing yield quality, profits and reducing environmental impacts. High investment costs and the need to maximize land productivity encourage the use of technology-based production methods in fruit and vegetable production (horticultural crops). These methods can lead to more precise and efficient management. Horticultural products are highly perishable, of high value and varying quality, and often require individual treatment of the plants in order to manage this variability effectively [1]. However, high-resolution spatial information capable of resolving individual trees is crucial to derive accurate management decisions. Satellite images often lack this detail and manual assessments are too labor-intensive. Unmanned aerial vehicles (UAV) have emerged as an interesting alternative to monitor orchards over the past decade.

UAVs have been used in orchards for a variety of purposes. Typically, they have been used to gain insights into orchard health and status. Since UAVs can be mounted with different sensors; UAV remote sensing for orchard assessment has been investigated with RGB cameras [2–5], multi- [6,7] and hyperspectral cameras [8,9], thermal cameras [6,10] and LiDAR sensors [11–13]. Point clouds derived from these data, using photogrammetry or time-of-flight information, have been used to obtain three-dimensional surface models of the orchards. They have been used recently to delineate tree structures in orchards [14–17], detect line crops for automatic navigation [18,19], estimate yield potentials per tree [20] and identify pest identification [21]. Apart from UAV remote sensing, UAVs can also be applied actively, for example, for spraying plant protection agents [22,23].

Fruit size estimation using 2D images is limited, as it requires calibration targets or combination with distance data, increasing complexity. Three-dimensional sensing techniques, such as LiDAR, RGB-D and multi-view stereo vision mounted on terrestrial platforms have proven to be more effective. For example, LiDAR has been used to estimate diameters and sizes of oranges and apples, with $R^2 = 0.63$ and $R^2 = 0.67$, respectively [24,25]. Structured light has shown potential in 3D analyses of grapes, with an $R^2$ of 0.70–0.91 in the prediction of cluster size [26]. Hacking et al. [27], used RGB-D to determine the volume and mass in grapes. Yu et al. [28] used 3D sphere fitting to estimate the position and size of pomegranates, achieving a root mean square error of 2.35 mm and $R^2$ of 0.82, while the position error was less than 5 mm. In parallel, stereoscopic vision was applied to harvesting robots to determine bunch volumes, with errors less than 17 mm and 19 mm in the estimation of height and maximum, respectively [29]. Furthermore, the multi-view stereo vision and the structure-from-motion reconstruction method were used to estimate diameters and volumes of fruits, such as apples and grapes, providing differences of about 2 mm with respect to manual measurements [30,31].

Although these methods provide accurate results for crop load estimation, they are often time-consuming, labor- and cost-intensive. UAVs stand out for their flexibility, low cost and repeatability of results [32]. The sensors they carry generate a large amount of data, mainly through images or video [33,34]. In this context, combining images from UAVs with deep learning techniques offers a more cost-effective solution than traditional methods based on human labor [35]. Nevertheless, the detection of fruit on tree tops can be difficult, especially when the fruit is obscured by other fruit or leaves. In these cases, additional data are needed to improve accuracy, and simple contour frameworks often fail. Apolo-Apolo et al. [36], exploited UAV images and the Faster R-CNN algorithm to detect oranges, whereas RGB videos acquired from a UAV were used to extract grape size, revealing an accuracy of 79.5% with the spatial embedding method and 44.6% with a YOLACT algorithm [37]. With timely information for location and status of the ripe fruits



in orchards available, more precise harvesting techniques that reduce fruit spoilage can be implemented [38].

This research aims for improving the detection of ripe apple fruits with a specialized UAV flight setting with a low flight altitude using an oblique-view camera perspective as introduced in Hobart et al. [16]. We hope to improve the signal-to-noise ratio of the obtained data by orientating the flight route with the tree rows, which should give an enhanced perspective view of the depicted tree crown area, and enhance the visibility of apple fruits. Specifically, this study explores an automated method for apple identification and yield estimation directly from a UAV-based photogrammetrical RGB point cloud. This approach has been rarely employed, to our knowledge. However, since three-dimensional point clouds are becoming increasingly common for orchard analysis, using these data for the direct estimation of orchard parameters should not remain unexploited as an information source. This also applies to apple mass and yield prediction directly from photogrammetrical RGB point clouds.

To explore the best flight setting for a newly developed automated apple detection and yield estimation procedure, UAV datasets from two different flights were compared with manually obtained reference data to test the following hypotheses.

1.  Ripe apple location and size of the 'Jonaprince' and 'Gala' variety can be automatically derived from a UAV-based, photogrammetric RGB point cloud.
2.  The estimated volume of the apples found can be used to estimate harvest mass for the orchard.
3.  The flight altitude of the UAV has an influence on the quality of the harvest mass estimation.

## 2. Materials and Methods

### 2.1. Test Site

Measurement campaigns were conducted on 4 September 2018 and 7 September 2020 in Marquardt at the Fieldlab for Digital Agriculture of the Leibniz Institute for Agricultural Engineering and Bioeconomy (ATB), Germany ($52°27'59''$ N, $12°57'29''$ E). The test site consisted of 1334 apple trees over an area of 0.5 ha. As shown in Figure 1, four rows were chosen for each UAV flight (rows 7–10 for 2018; rows 2–5 for 2020), each with 112 trees spanning a combined area of 1200 m². The orchard was planted with 'Gala' ($n = 240$) and 'Jonaprince' ($n = 180$) varieties, as well as pollinator trees of the varieties 'Red Sentinel' ($n = 16$) and 'Red Idared' ($n = 12$). The pollinator trees were homogeneously distributed over the orchard. Measurements were taken shortly before harvest at 128 and 133 days as well as at 132 and 133 days after full bloom, for 'Gala' and 'Jonaprince' trees in 2018 and 2020, respectively. In September, the ripe apples of the named varieties showed a bright red color, which enabled later RGB based analyses. The trees were regularly pruned to slender spindle trees. The row distance was 5 m and distance between trees was 0.75 m, constituting a continuous fruit wall. The tree wall had a height of about 3.0 m and a width of about 1 m. The rows were separated into 14 blocks of 8 trees each.

### 2.2. UAV Measurements

Flight campaigns were conducted at fruit ripening, a few days before harvest. The flight missions were conducted with an octocopter (CiS GmbH, Rostock, Germany) carrying a consumer-grade RGB camera ($\alpha$-6000, Sony, Tokio, Japan). The system had a takeoff weight of less than 2 kg and was capable of flight times of up to 30 min. To reduce blurring and obtain a fixed camera angle, a two-axis gimbal was used. The camera had the following specifications: 24.7.5 megapixel APS-C chip with a sensor size of 23.6 mm $\times$ 15.8 mm; the resulting pixel pitch was 3.9 μm. The focal length used for all campaigns was 16 mm, the

ISO was kept at a constant 400 to minimize noise effects and the aperture and exposure time were adapted to light conditions.

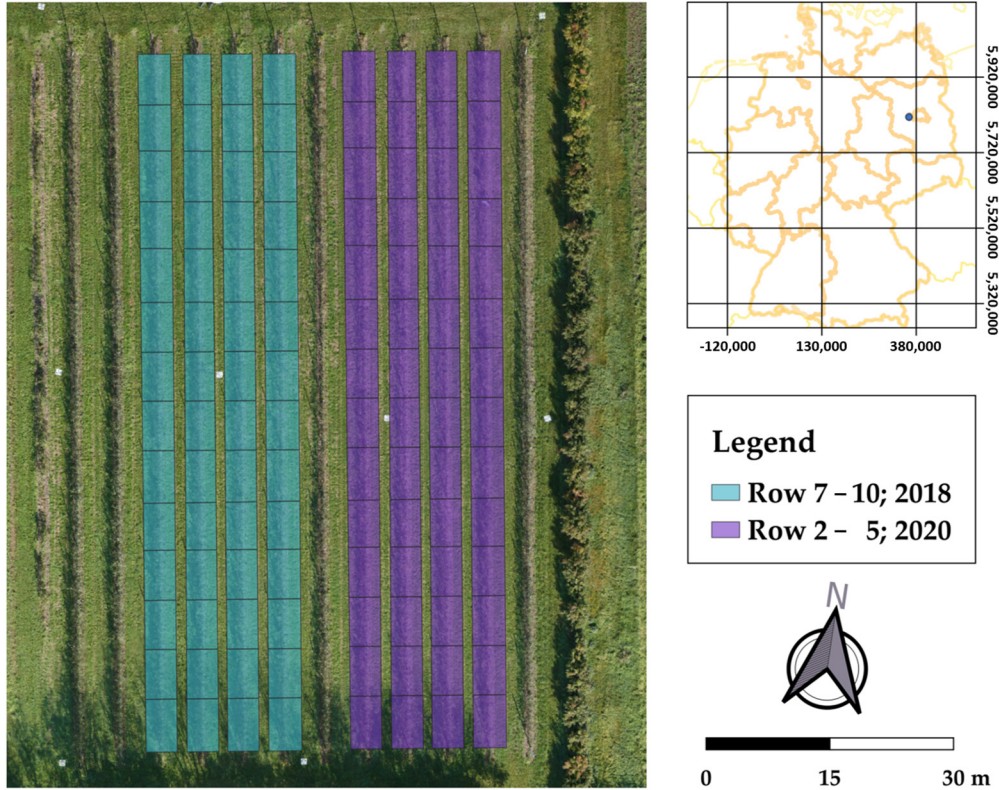

**Figure 1.** Test site as ortho image superimposed with blocks of apple tree rows investigated in 2018 (blue) and 2020 (purple). Also, the location in Germany is given in the upper right corner.

For each measurement date, multiple flights were conducted at two different altitudes. The tested altitudes were 7.5 m and 10 m for an oblique perspective on the neighboring tree wall. For the oblique perspective, one campaign consisted of three flights [16]. One contour flight was planned along the outer border of the tested area to give an overview of the test site. Two detail flights were conducted at the same altitude, following the path along the tree rows and capturing the tree crowns of the neighboring row to the left and to the right. In this way, relevant information from the tree crowns was captured with an unobstructed view and a small sampling distance (SD). The SD on the tree wall surface, comparable to the ground sampling distance but calculated for the vertical tree wall surface at a tree height of 1.5 m above ground level, is shown in Table 1 for different detail flight altitudes along with the given camera angle.

**Table 1.** Captured and aligned photographs per measurement date for different flight settings and resulting image parameters.

| Date/Parameter | 7.5 m | 10 m |
|:---:|:---:|:---:|
| 04.09.2018 | 407/406 | 342/342 |
| 07.09.2020 | 1802/1802 | 1222/1200 |
| Velocity (m/s) | 0.4 | 0.5 |
| Vertical angle (°) | 53 | 46 |
| Sampling dist. (mm) | 1.9 | 2.4 |

The flights were conducted at different velocities (0.4 m·s$^{-1}$ and 0.5 m·s$^{-1}$, for 7.5 m and 10 m, respectively) to keep the forward overlap close to 90% and 96%, for 2018 and 2020,



respectively. The difference between the two dates derived from a technological update of the camera storage device. That allowed a higher capturing frequency in 2020 compared to 2018. The number of captured images from these flights is shown in Table 1. For georeferencing and combining the several point clouds, marker plates were visibly placed within the orchard, and the coordinates of each plate were recorded with a GNSS-RTK system (HIPer Pro, Topcon, Tokio, Japan).

### *2.3. Image Data Processing*

### 2.3.1. Point Cloud Calculation

Captured RGB images were transformed from ARW to 16-bit .tiff file format using an open-source image processing software tool (Gábor Horváth and RawTherapee development team, version 5.8, Budapest, Hungary). For the imagery, 3D point clouds were calculated photogrammetrically for datasets from each flight setting using a specialized software tool (Metashape Professional, version 1.8.4, Agisoft LLC, Sankt Petersburg, Russia). The alignment process was conducted with high accuracy and georeference plates, GPS-RTK located in the field, were used to enhance the quality of the alignment and to set the coordinate system to ETRS89 UTM zone 33N (EPSG 25833). To exclude matching points with high inaccuracies and achieve an even better alignment, a gradual selection was conducted for all the point clouds. Points from sparse clouds were excluded if they exceeded the values of 0.1 for reprojection error, 50 for reconstruction uncertainty, 20 for projection accuracy. After each step, a photo alignment optimization was performed, leading to higher error margins in the other categories. Therefore, the threshold of 0.1 for the reprojection error was set a second time at the end. The selection was conducted stepwise, and photograph alignment was optimized after each step. If photographs were excluded in this process, the selection was conducted with higher error margins to prevent information loss. The exact values are shown in Table A1 (Appendix A). The optimized sparse point clouds were used as a basis to calculate separate dense clouds for the different flight settings with high accuracy and mild depth filtering.

### 2.3.2. Apple Identification and Volume Estimation

The process of apple identification from the RGB point clouds can be seen in the flowchart in Figure 2. Tree and ground surface points were separated from each other using the cloth simulation filter [39] in the open-source point cloud processing CloudCompare software (version 2.12, GNU GPL). The used rigidness was set to 2 (scenes: relief), the slope processing was turned on and the cloth resolution and max iterations were set to 0.5 and 600, respectively. Points below the ground surface were deleted manually to avoid unstable classification results. With the geoinformation system software QGIS (version 3.34.4) [40], polygon objects were created to select tree blocks and tree rows in the measurement orchard. This was used to cut the created point clouds into 14 tree blocks per row with the statistical programming language R (version 4.4.0) [41]. Using the package 'lidR' [42,43], the function clip was used to cut the tree point clouds into sub clouds showing blocks of 8 trees each. Using the 'grDevices' package, a transformation from RGB point values to the hue, saturation and value color space was conducted. This gave the opportunity to filter for points with a hue value between 0.00 and 0.04 or between 0.96 and 1.00, excluding all points except red ones. To remove the white points, further points were extracted from the red sub cloud if they showed values (brightness) higher than 0.75 or saturation of less than 0.25.

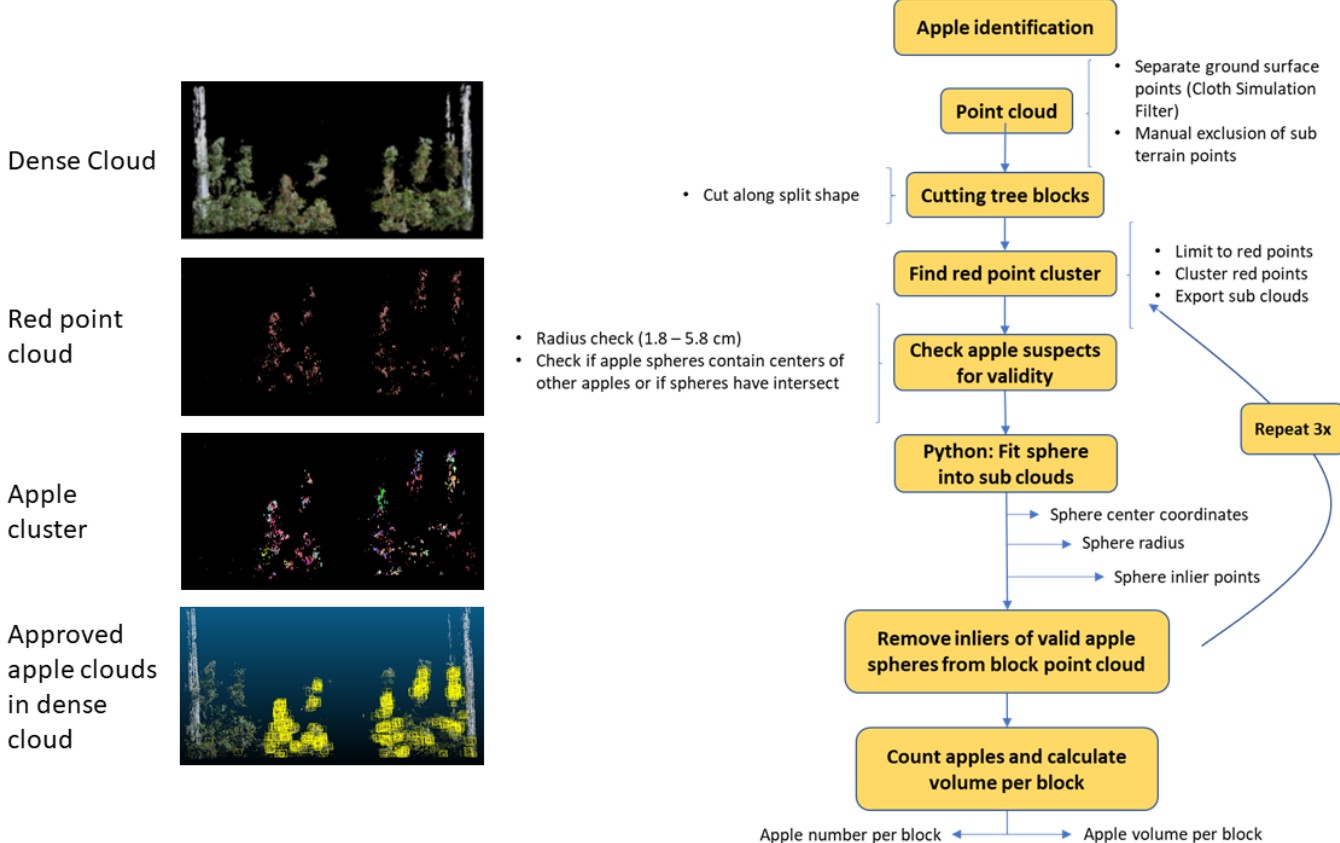

**Figure 2.** Flowchart of apple identification from RGB point clouds.

To find positions of suspected apples, the sub cloud was clustered using the DBSCAN algorithm [44] from the 'dbscan' package [45,46], which estimates a point density around each cloud point and uses a maximum distance to define cluster inliers or outliers of neighboring points, according to a maximum distance set by the user. Border points of each cluster were considered cluster inliers in all cases. To establish the best performing model, all combinations of parameter ranges were tested on a test block over all dense clouds. The ranges in which the parameters were set were 6, 8 and 10 for the minimal number of points to form a fruit cluster, the epsilon value, giving the minimal distance to separate two clusters, of 0.01 m, 0.012 m, 0.015 m, 0.018 m and 0.02 m. Tested and set parameter values are summarized in Table 2.

**Table 2.** Parameter set overview for fruit clustering and identification algorithm.

| Parameter | Tested Values in Optimization | Set Value |
|---|---|---|
| Cluster distance (Epsilon) | 0.010 m, 0.012 m, 0.015 m, 0.018 m, 0.020 m | 0.015 m |
| Border points considered inlier | - | Yes |
| Minimum number of points per cluster | 6, 8, 10 | 6 |
| Cluster radius range | - | 0.018–0.058 m |
| Red color value range | - | 0.00–0.04 and 0.96–1.0 |

The best performing model was then used to separate the red points into apple suspect clusters. If a cluster had an extent of more than the doubled average apple diameter, it was split up into sub clusters using the Hartigan and Wong algorithm from the Kmeans function algorithm [47]. All clusters were further processed using the library pyntcloud [48]

for general data handling and pyransac3d [49] in Python (version 3.10.4, Python Software Foundation, Beaverton, OR, USA). The latter library was used to create a three dimensional sphere using the function 'pyrsc.Sphere' and fit it to each point cluster using the PGP2X algorithm [50] from the 'sph.fit' function. The center, radius as well as inlier points of each sphere were retained. In the next step, spheres outside the radius margins of 1.8 to 5.8 cm were excluded. The remaining spheres were checked for their overlap. From the radii and center points, apple volumes and overlap between spheres were calculated. In case of overlaps larger than 10%, the sphere with a difference higher than r = 4.25 cm was deleted. The remaining spheres were stored as verified apple spheres. Then, all inlier points of the stored apple spheres were deleted from the original block cloud and the loop was started again from the process of clustering. This process was repeated four times to find missed apples in the remaining red points.

## 3. Results

In general, it was possible to delineate three-dimensional point cloud models from data of both flight settings and for both years. The model was able to automatically identify apples in the provided dense point cloud and estimated the volume of each. An example is shown in Figure 3, where bounding boxes of the located apples for one block in the 10 m altitude 2018 dense cloud are depicted. The block of eight trees was cut from the orchard point cloud and the estimated apple bounding boxes show apple sizes in realistic areas of seven of the eight trees. The tree on the right-hand side did not have any apples because it was a pollinator tree. While many apples were identified, some fruits were not well represented in the dense point cloud or not found. This can be considered to some extent due to overlapping of leafage or to a clustering of multiple apples to which a sphere could not be fitted. The model will therefore most likely underestimate the number of apples in the orchard.

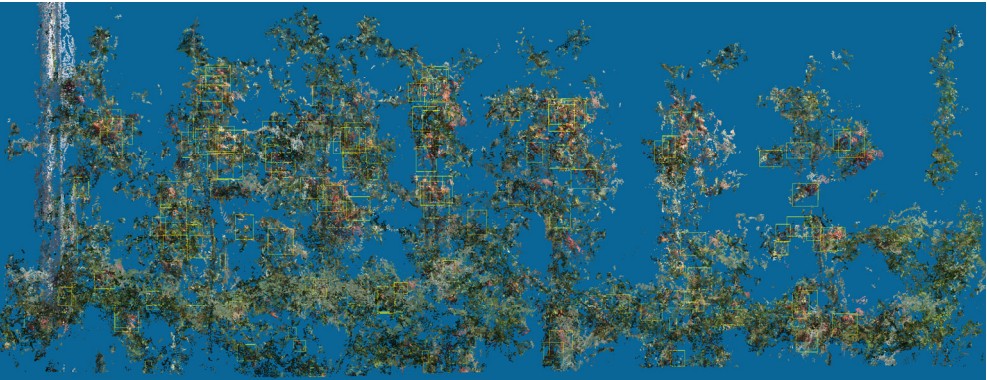

**Figure 3.** One block from 10 m dense cloud with bounding boxes (yellow outlines) of approved apple positions.

### 3.1. Model Optimization

There are many different parameters that can influence the model performance. In order to run the model with the best parameter set possible, the difference in the estimated apple number of a test block and its reference was calculated for different parameter combinations. At first, parameters were tested manually to find rough values, then, in a second step, the parameters were adjusted, defining test ranges for all parameters. Three of the four provided dense clouds were analyzed with every possible combination of these parameter values (as described in Section 2.3). Excluded was the 7.5 m cloud from 2020, which produced no results, because of multiple superimposed point clouds leading to a

high level of shifted points. The parameters were kept steady for all four repetitions of the model.

In Figure 4, a comparison between model estimations and the reference apple number for one test block of the orchard for all point clouds is shown over changing parameter sets. Optimally, the difference with the reference would be zero. Results show that for this one block, the general performance is highly dependent on the flight setting. This seems realistic because of the difference in point density leading to a different cluster distribution. The point clouds from a 10 m altitude flight had smaller errors for all the parameter sets. In 2018, this point cloud underestimated and in 2020, the same setting overestimated, the apple number in the tested block. From one 7.5 m flight, a highly overestimated apple number was found. After the created apple identification model was trained on one block for each year, the mean performance of all dense clouds showed the best performance for a minimal six points per cluster and a cluster separating distance of epsilon = 1.5 cm.

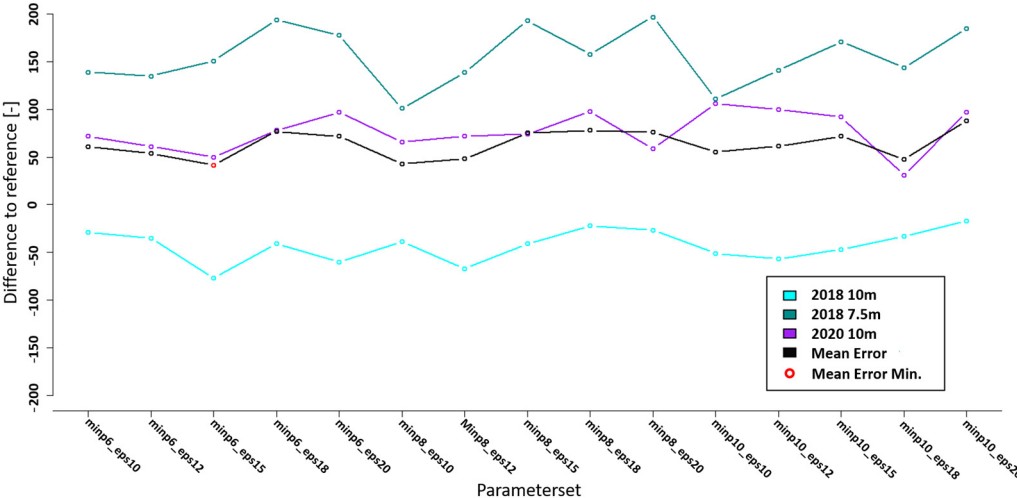

**Figure 4.** Performance test for apple identification model for different parameter sets, minimum points per cluster (min), and distance to distinguish two clusters (eps) for all point clouds.

The model output included the number of found apples, the position, as well as the radius of the fitted sphere of the apple. To calculate the yield mass per block, the mean density for both apple cultivars was calculated based on >100 ripe samples per cultivar, as shown in Figure 5. In this way, a density factor per breed was estimated, which was 1459.73 kg/m$^3$ and 1465.39 kg/m$^3$ for 'Gala' and 'Jonaprince', respectively. This factor was later multiplied by the calculated sphere volumes.

In Figure 6, located apples are depicted superimposed on the ortho image of the test site. Some blocks in 2020 were empty because they were already harvested, with almost no apples left on the trees. When analyzing the 7.5 m point cloud, the model found more apples compared to the point cloud generated from the 10 m altitude. However, these apple localizations showed a higher dispersion in position across the tree rows, which gives the impression of a less accurate point cloud.

Apple estimations were, in addition, worse for both years and flight settings were worse along the last flown tree row, as shown in Figures 7 and 8. Row 10 in 2018 showed coefficients of determination ($R^2$) of 0.00 and 0.03 for the 7.5 m and 10 m altitude flights, respectively. In comparison, the mean $R^2$ of models fitted on the basis of the other three individual rows was higher, with 0.52 for 7.5 m altitude data and 0.54 for 10 m altitude data.

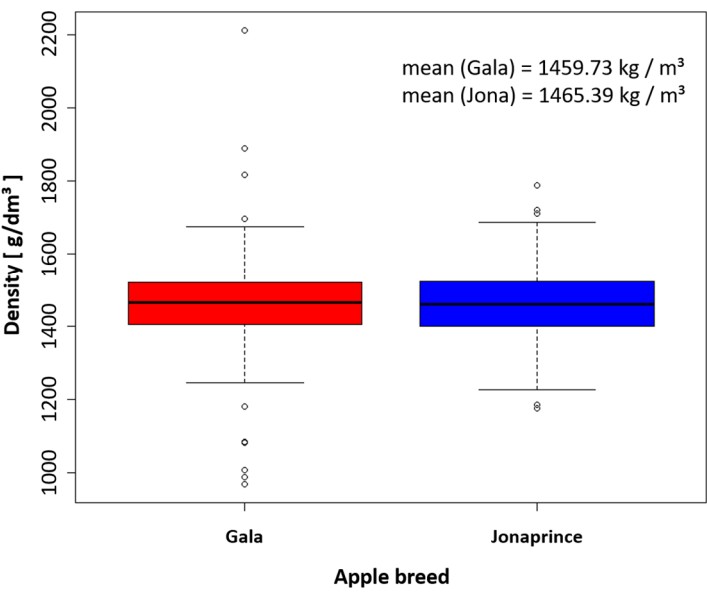

**Figure 5.** Mean apple density per breed.

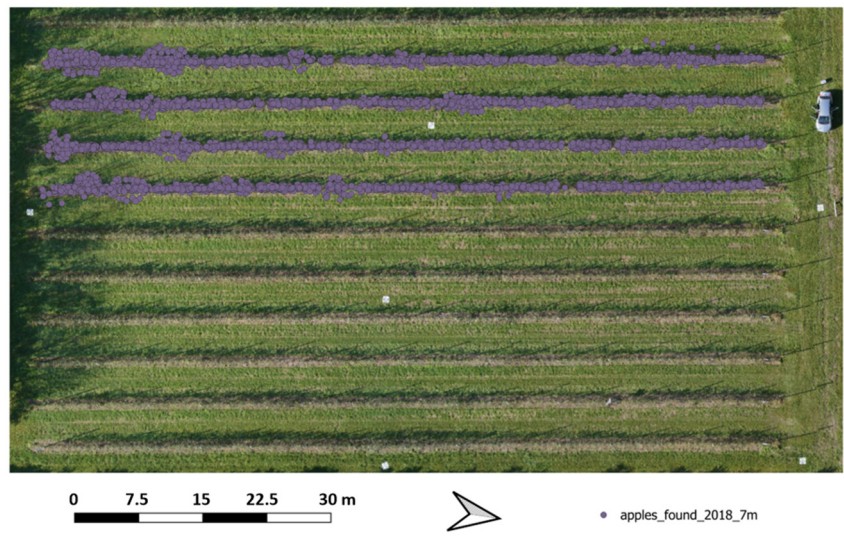

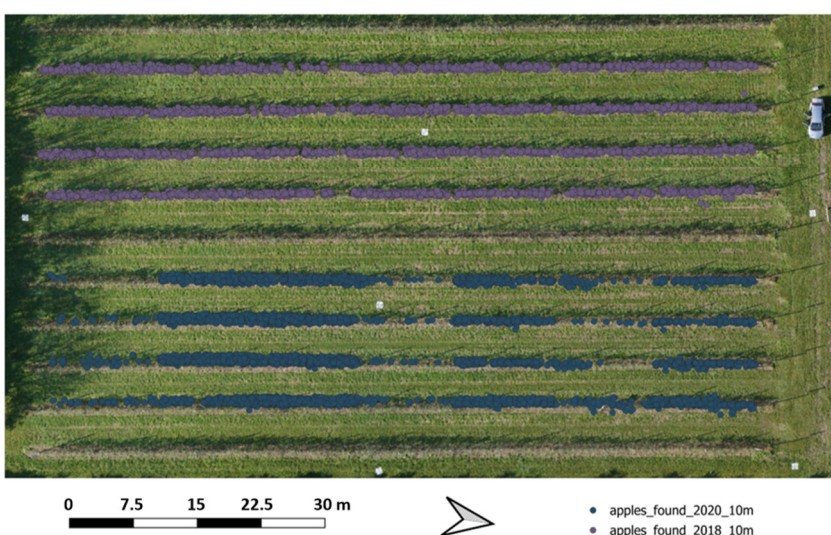

**Figure 6.** Computed apple centers for the 7.5 m (**top**) and 10 m (**bottom**) flights.

The same applies to Row 5 in 2020, which had an $R^2$ of 0.06 for the 10 m altitude dataset. Compared to the mean coefficient of determination of 0.5 for the other three rows, this value is worse as well. This effect was found for the last flown tree row for both years and is explained by the fact that the outer canopy site is captured on fewer images. Therefore, the data basis for the delineation process is diminished. This effect was found to be only one-sided, which was probably caused by a minor offset in flight coordinates to the east. For that reason, these rows were excluded from further steps of the analysis, and models were fitted on the basis of the remaining apple tree rows.

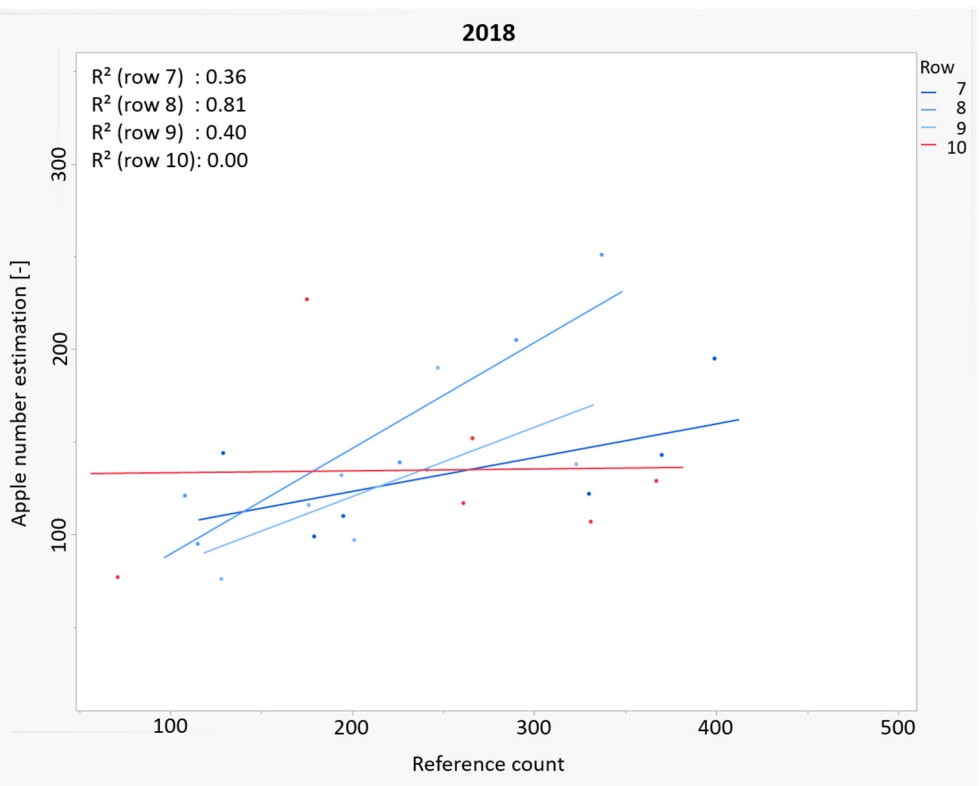

**Figure 7.** Model (blue or red line) performance for apple identification estimation from a 7.5 m altitude perspective per row (row 7, 8 and 9: blue color; row 10: red color).

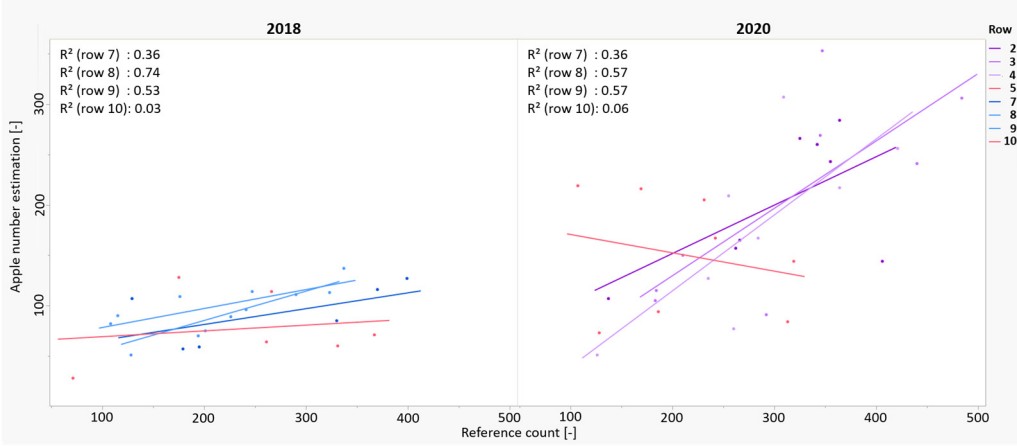

**Figure 8.** Model (blue, red or purple line) performance for estimated apple identification from a 10 m altitude perspective per row for both years (row 2, 3 and 4: purple color; row 7, 8 and 9: blue color; row 5 and 10: red color).

### 3.2. Apple Identification

For the remaining tree rows, block-wise apple count models were fitted. At an altitude of 7.5 m, a model fit with an $R^2$ of around 0.41 was achieved. For the datasets recorded from an altitude of 10 m, fits with an $R^2$ of 0.40 and 0.53 were achieved for the 'Gala' and 'Jonaprince' blocks, respectively, as shown in Figure 9.

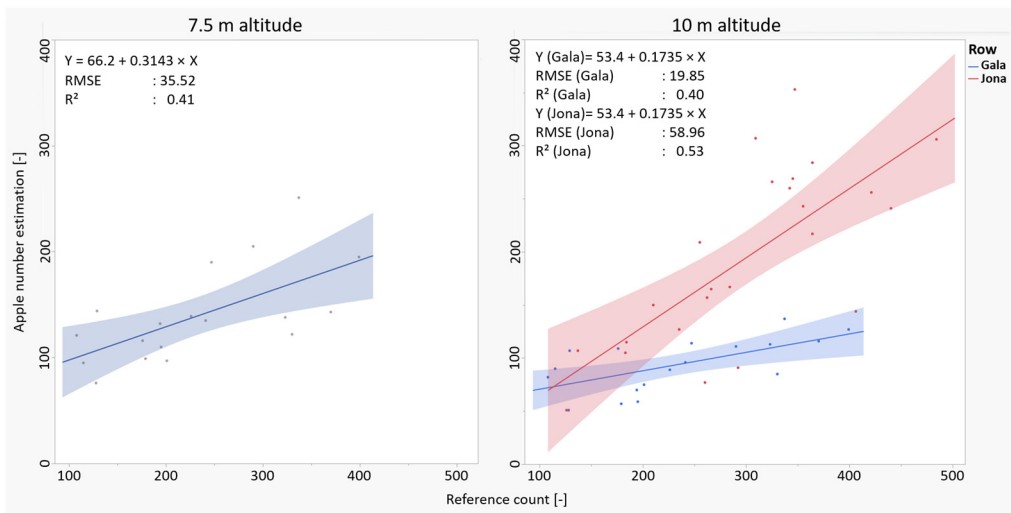

**Figure 9.** Overall model (blue or red line) performance and confidence interval (blue or red area) on three remaining apple rows for estimated apple identification from both altitudes for tree blocks of the Gala (blue) and Jona (red) variety.

### 3.3. Yield Estimation

As with the apple identification models, model adjustments were made for the expected apple harvest mass. For this purpose, the estimated volume of all apples in a block was multiplied by the average apple density of the corresponding variety. The adjustments show $R^2$ of around 0.56 and 0.76 for the data from the 7.5 m and 10 m flight altitudes, respectively, as shown in Figure 10. Thus, a very high proportion of the measured variance can be represented by the adjusted models using the derived apple volumes in conjunction with a mean apple density per breed.

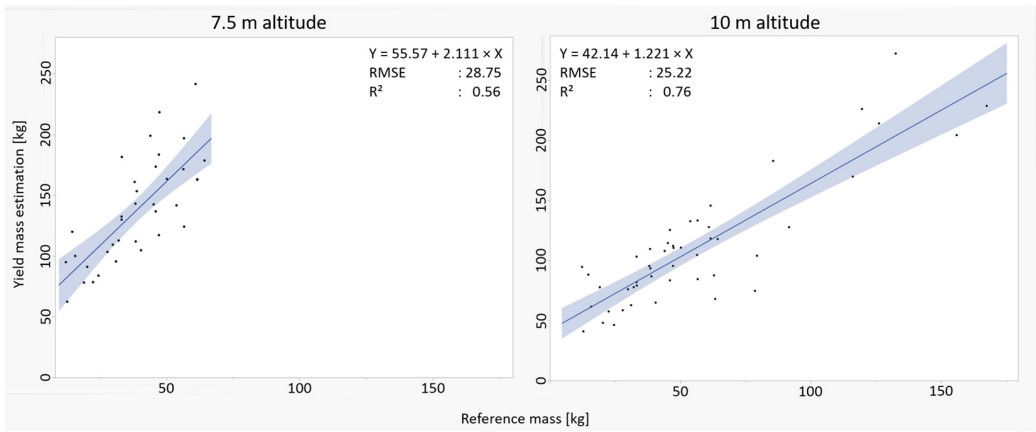

**Figure 10.** Overall model (blue line) performance and confidence interval (blue area) on three remaining apple rows for apple mass estimation from both years for separate flight altitudes.

When the model was applied to the tested seasons, a map of the estimated yield mass per block could be calculated. These estimated yield maps are shown in Figure 11 for the 7.5 m dataset in 2018, Figure 12 for the 10 m dataset of the same year and in Figure 13 for

the 10 m dataset from 2020. In 2018, the same trees were surveyed on the same day with flights at altitudes of 7.5 m and 10 m. However, the scale for the estimated yield based on the 7.5 m point cloud analysis was about 20 kg higher at a minimum and 40 kg higher at a maximum compared to the 10 m flight-based findings. The same patterns can be found by comparing both model results. This underlines the idea of a more dispersed point cloud dataset from the 7.5 m altitude flight.

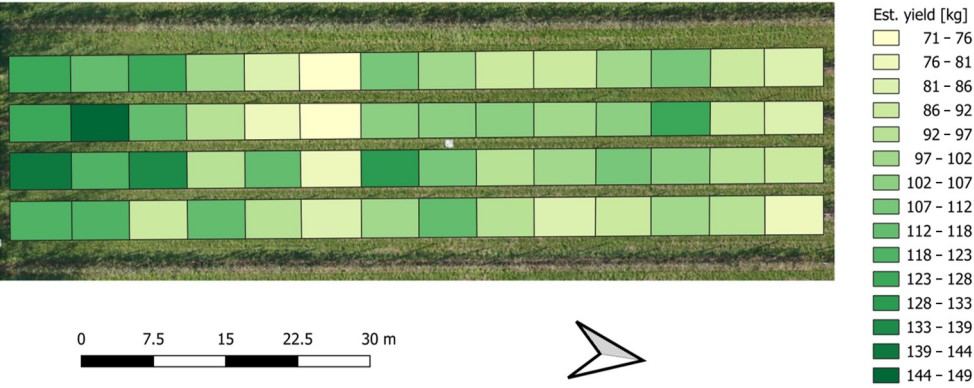

**Figure 11.** Apple yield estimation per block in 2018 for 7.5 m point cloud.

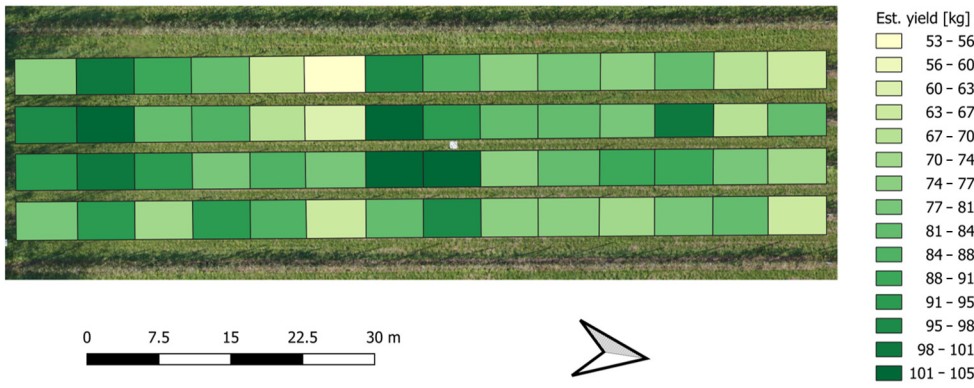

**Figure 12.** Apple yield estimation per block in 2018 for 10 m point cloud.

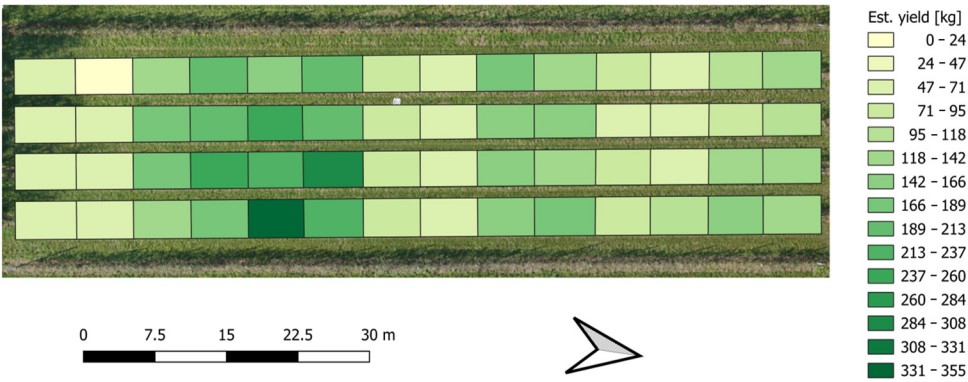

**Figure 13.** Apple yield estimation per block in 2020 for 10 m point cloud.

The 10 m point cloud from 2020 showed a much wider range in the yield mass distribution compared to the 2018 dataset. Small yield estimations are reasonable coming from the partially harvested field before data were captured, leaving some blocks empty of apples. For the higher yield estimations, the range shows about tripled estimated yield values compared to the block estimations from 2018. This finding is reasonable, because reference yield values for 2020 were higher and ranged from 35 kg to 125 kg per block for

2018 compared to 63 kg to 168 kg per block for 2020. However, the estimations for 2020 were still too high, and two blocks showed estimated yields above 255 kg.

## 4. Discussion

For modern apple orchard management, it is appealing to rely on remote sensing technologies for yield estimation and fruit identification. Given the large volumes of data generated, automated analysis methods become essential to extract parameters for site-specific decision-making. However, most existing approaches use two-dimensional data to delineate yield and fruit locations.

When it comes to fruit detection and localization, Zhou et al. [51] described a method before machine learning became a popular approach in precision horticulture. They used pixel-based thresholds of spectral indices using decision trees applied to a ground-based two-dimensional image dataset. The used color information for an apple fruit recognition system gave insights into an estimated yield shortly before harvest and early in the year. The fruit-counting model, when compared to manually taken apple counts, reached $R^2$ values from 0.8 to 0.85 close before harvest, and for yield estimation, from 0.57 to 0.71. In comparison, $R^2$ values presented in this study were much weaker for apple counts, at 0.4 to 0.5. We argue that a UAV-based image dataset provides less spatial resolution and a weaker angle to gain information in the tree canopy compared to ground-based image datasets. However, for the apple yield estimation, $R^2$ values ranged from 0.56 to 0.76, which is highly comparable with the findings from Zhou et al. [51] and shows that a yield estimation in an apple orchard can be derived from UAV 3D point clouds.

Fruit detection nowadays is mainly researched from two-dimensional RGB [52–54] or RGB-D [38,55–57] data. A common and successful approach works with artificial intelligence (AI)-supported object-based image analysis. Often, apple detection models were developed with the focus on automatic harvesting from robotic systems. They were optimized to work accurately and fast on two-dimensional, ground-based image data input. These approaches reach very high accuracies and fast processing speeds. However, they require a fully automated orchard or laborious manual data collection. A bit different is the setting in the publication from Apolo-Apolo et al. [58], in which a region-convolutional neural network was trained to detect apples in an orthomosaic of a whole orchard from UAV-based RGB imagery. The data were captured at an altitude of 10 m, and analyses led to an $R^2$ value of 0.80 comparing the estimated number of apples per tree to in situ reference counts. Another approach to orchard-scale yield estimation uses indirect correlations to yield mass, with certain parameters derived from UAV imagery, e.g., spectral index [59] or tree structural parameters in combination with breed and management differences [60].

The study presented here explores a method of automating apple identification and yield estimation directly from an orchard-wide photogrammetrical RGB point cloud. Zine et al. [61] also directly used three-dimensional RGB-point clouds for apple identification. From ground-based, manually collected imagery close to the 32 researched trees, apples were detected, counted, and assigned to the individual trees in a dense apple orchard. Similarly to our study, apple detection was accomplished with a color-based threshold after an HSV transformation of the RGB data. The remaining points were converted to volumetric form and stored with its bounding box as well as center position. The center point was considered as the apple location. The apple detection showed a recall rate of 74–90%, which is better than in our study. We argue that the apple identification provided here is worse due to the UAV-based data acquisition method. The reduced resolution due to the distance and the more unstable photo setting leads to less accurate point clouds, which cannot provide the same results as those based on ground-based images directly next to the trees. In the work of Tsoulias, Saha and Zude-Sasse [12] a ground-based LiDAR point

cloud was used to estimate the number of fruits and their sizes from 12 apple trees. Fruit point clusters were separated from the tree cloud using a curvature value and the response intensity of the LiDAR signal. As the point cloud quality of the test trees was high, fruit number and fruit size estimations showed a superior $R^2$ of 0.99 and $R^2$ of 0.98 compared to the study provided here. Due to the more expensive sensor and the ground-based data recording approach, the comparability with the practical approach, as shown in the study presented here, is low. However, the idea of direct fruit positions and structure extraction from a three-dimensional point cloud is the same.

As was shown in comparison to Zine et al. [61] and Tsoulias, Saha and Zude-Sasse [12], the approach relies on the quality of the provided point cloud model. That is influenced by the UAV flight settings for image collection. To enhance the resolution, two low altitude flight settings for the image capture process were tested in the provided study. In this way, highly detailed, most accurate point cloud models should be provided for the analysis procedure. From apple identification results with the test block, shown in Figure 4, the findings indicate higher errors for the model based on the 7.5 m flight setting. As shown in Figure 9, the 7.5 m point cloud led to results of similar quality but increased estimates of apples found compared to point cloud results generated from 10 m altitude imagery. These findings show a stronger dispersion in the position across the tree rows, as shown in Figure 6, giving the impression of a less accurate point cloud. This would also explain the additional apples found, when apples are depicted with a higher dispersion leading to false positives or larger clusters that are split up in the model processing. The idea of a less accurate point cloud basis for the 7.5 m dataset is further supported by Table A1 (Appendix A), which shows a necessity for higher reprojection error margins compared to the 10 m point cloud creation process. In addition, as shown in Figure 10, apple mass estimation led to weaker model results for the whole orchard when the 7.5 m point cloud ($R^2$ of 0.56) was compared to findings from the 10 m point cloud data ($R^2$ of 0.76). In summary, it can be stated that according to the results presented here, a point cloud generated from 10 m altitude imagery leads to less dispersed results in apple fruit position and better apple mass estimation compared to that generated from the 7.5 m point cloud.

Analyses of photogrammetrical point clouds of whole orchards for fruit mass or yield prediction, to our knowledge, have also been rarely researched. Torres-Sanchez et al. [62] used UAV-based photogrammetrical point clouds and color analysis from imagery taken from oblique flights at altitudes of 10 m and 15 m for the detection of grape clusters. In an unsupervised and automated workflow in red grapevine varieties, an algorithm was developed that models the harvest weight from the projected area of the points classified as grapes on vines. The model achieved $R^2$ values higher than 0.75 when leaves were removed from both sides of the plant row, to better capture the grapes in the image dataset. In their discussion, the occlusion by leaves, which hides fruits and hinders detection, was presented as a major problem. Also, Géne-Mola et al. [30] researched the influence of occlusion on ground-based RGB structures from motion point clouds. From these, apple fruit locations and diameters were estimated depending on the degree of occlusion. It was shown that diameter estimations for a least squares sphere fitting approach, similar to the algorithm presented here, show smaller RMSE and better $R^2$ values with higher visibility of the fruit. It is reasonable that leaf occlusion also lowered our apple estimation results, as it applies also to apple fruits and grapevine fruit clusters. In the case of an orchard with very dense apple fruit walls, this could affect the functionality of the algorithm. However, we tried to limit this problem by using a low-altitude, oblique view perspective rather than using nadir perspective from the UAV platform.

One more limiting factor for fruit detection in orchards based on RGB data is the light condition during data capture. This was, e.g., described by Sengupta and Lee [63], who

developed an approach to automatically identify citrus fruits. According to them, RGB imagery of fruits in orchards face difficulties, e.g., with different illuminations, partial occlusion, the presence of highly saturated regions, different orientations, very diverse backgrounds, and the presence of shadow regions. They emphasize that illumination has a strong influence on automatic fruit identification. These influences also play a role in the creation of the point clouds on which the approach shown here is based. Although the influence of ambient light was not investigated here, it should be addressed in subsequent studies. This is the only way to ensure stable RGB data-based fruit detection from photogrammetric point clouds.

## 5. Conclusions

In this study, a model for automatic apple fruit detection with dense 3D point clouds derived from UAV-based low-budget RGB imagery at different flight altitudes was developed. The algorithm was tested in a field study in an apple orchard in Brandenburg, Germany, with data captured in two seasons. The automated algorithm showed its best performance for the input parameters with a minimum of six points per fruit cluster and a separation distance of 1.5 cm between two clusters (epsilon). It was able to detect and locate apple fruits in three of the four provided point clouds. The fourth dataset, obtained from a flight at the 7.5 m altitude in 2020, caused the algorithm to show empty results, as the provided dense clouds contained multiple, superimposed point clouds that caused shifted points, which appear as a high level of noise. The completely automated algorithm was able to split the three-dimensional orchard point cloud into zones of interest along a provided polygon vector, extracted points of ripe apples according to their hue value after RGB to HSV transformation, clustered the fruits, and sub-clustered groups of fruits. Furthermore, the algorithm fitted a sphere into each suspected apple fruit and checked for reasonable radius values before the location and volume were stored. The developed analytical procedure was therefore capable of automatically delineating the location and volume of ripe apple fruits of the 'Jonaprince' and 'Gala' varieties from the point clouds.

Yield estimations showed $R^2$ values of 0.71 and 0.56 for the 10 m and 7.5 m flight altitudes, respectively, in comparison to yield masses calculated from orchard block harvesting. Delineated apple fruit numbers and yield masses were shown as maps in a site-specific distribution for the test orchard for both years. In this way, the provided method is capable of estimating apple fruit volumes, which can be used to delineate site-specific yield mass estimations for the orchard. The approach was tested for two low-altitude flight settings for the image data capture process. It was shown that point clouds generated from a 10 m altitude image basis lead to less dispersed results in apple fruit position and better apple mass estimation compared to those from the 7.5 m point cloud basis.

Thanks to advances in UAV technology, the use of three-dimensional point clouds is becoming increasingly popular in modern orchard management. Thus, these datasets should be further explored to delineate important tree-structural and growth parameters and thus lead to improved orchard management. In future research, neural networks could be trained with labeled fruits from the algorithm presented here or manually assigned labels in a 3D point cloud environment. These neural networks could enhance accuracy and processing speed for apple fruit detection from three-dimensional point clouds, although this approach will require more data and computational resources than the method presented in this study. Moreover, findings from the research field of automatic fruit shape completion could enhance volume estimations with the provided model.

**Author Contributions:** Conceptualization, M.H., M.S. and M.P.; Data curation, M.H. and M.K.; Formal analysis, M.H.; Funding acquisition, M.P. and M.S.; Investigation, M.H., M.P. and M.S.; Methodology, M.H., N.T., M.P. and M.S.; Project administration, M.P. and M.S.; Visualization, M.H.;

Writing—original draft, M.H.; Writing—review and editing, M.S., N.T. and C.W. All authors have read and agreed to the published version of the manuscript.

**Funding:** The research project on which this paper is based was funded by the German government, grant number 2814903915.

**Data Availability Statement:** The datasets generated for the study presented here are available from the corresponding author on reasonable request.

**Acknowledgments:** For the fundamental project support, we thank Ludwig Schrenk and his Team from CiS-GmbH Rostock; Mirko Bohte and Sergej Krukouski, who helped us regarding the UAV, for their expertise in flight planning and conduct as well as point cloud generation. We would also like to thank Antje Giebel, who always assisted in GIS, point clouds, and statistical tasks. Further, we want to thank Giles Young, who gave our expressions the final polish.

**Conflicts of Interest:** The authors declare no conflicts of interest.

## Appendix A

**Table A1.** Sparse cloud gradual selection details.

| Chunk | Year | Tool | Value | Points Remaining |
|---|---|---|---|---|
| 7 m | 2018 | Sparse point cloud | | 414,178 |
| | | reprojection error | 0.1 | 101,731 |
| | | reconstruction uncertainty | 50 | 91,603 |
| | | projection accuracy | 20 | 88,054 |
| | | reprojection error | 0.3 | **85,507** |
| | 2020 | Sparse point cloud | | 1,558,660 |
| | | reprojection error | 0.5 | 916,904 |
| | | reconstruction uncertainty | 50 | 686,592 |
| | | projection accuracy | 20 | 679,896 |
| | | reprojection error | 0.5 | **670,046** |
| 10 m | 2018 | Sparse point cloud | | 458,186 |
| | | reprojection error | 0.1 | 105,305 |
| | | reconstruction uncertainty | 50 | 102,221 |
| | | projection accuracy | 20 | 98,218 |
| | | reprojection error | 0.1 | **71,493** |
| | 2020 | Sparse point cloud | | 1,138,353 |
| | | reprojection error | 0.5 | 722,328 |
| | | reconstruction uncertainty | 60 | 671,906 |
| | | projection accuracy | 20 | 666,541 |
| | | reprojection error | 0.3 | **463,051** |

Red values indicate higher error margins in order to prevent image data loss in the dataset; bold point numbers indicate the end result of sparse point cloud thinning.

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
