# Peer review of "Fruit Detection and Yield Mass Estimation from a UAV Based RGB Dense Cloud for an Apple Orchard"

_drones, doi:10.3390/drones9010060_

Round 1

Reviewer 1 Report

Comments and Suggestions for Authors

The manuscript explores a UAV-based approach for fruit detection and yield estimation in apple orchards using photogrammetric RGB point clouds. Conducted in Brandenburg, Germany, the study tested UAV flights at 7.5 m and 10 m altitudes over two years to detect ripe apples and estimate their volume and mass. The automated algorithm identified apples based on color and structure, achieving better accuracy at 10 m altitude due to reduced noise and improved data coverage. While the models showed underestimation due to leaf occlusion, they demonstrated a strong correlation with yield mass per block when apple volume was combined with density values. The study highlights the potential of UAVs and photogrammetric point clouds for precise yield prediction in orchard management.

The topic of your manuscript is of very interest to the research community. However, the manuscript requires major revisions before my final judgment. Please address the following comments during the manuscript revision process:

1. There are too many keywords in the keywords list, you may delete some non-related keywords according to the contents of the manuscript, and just remain 3-5 key words.

2. Introduction section, for the beginning of part include a broader discussion of the role of precision agriculture in orchard management, highlighting its economic and environmental importance. Add references to recent advancements in 2D and 3D fruit detection using machine learning and deep learning techniques for fruit detection. Explore studies on yield estimation using UAVs in various crops, not just apples, to establish a broader foundation. Point clouds and photogrammetry should highlight the growing application of photogrammetry in agriculture and its potential advantages over LiDAR. It is suggested that the author consider rearranging the contents of the introduction section.

3. For the citation style and the reference, it is suggested that the author follow the style of the journal.

4. Line105-107 the varieties ‘Gala’ (240), ‘Jonaprince’ (180), ‘Red Sen-tinel’ (16) and ‘Red Idared’ (12). What is the number in parentheses after the apple variety name?

5. What is the BBCH apple growth stage scale? How tall are apple trees?, How wide is the canopy? and What about apple canopy shape?

6. Materials and Methods section, subheading 2.3 Image Data Processing. It is suggested that the author divide the methods into subheadings 2.3.1, 2.3.2 .... etc. for easier understanding. 

7. It is suggested that the author address the high noise levels in the 7.5 m dataset. Consider employing advanced noise reduction techniques or refining flight parameters to minimize occlusion caused by foliage.

8. All commercial software and products were used, adding manufacturer, city, country.

9. In the results section, it is suggested that the author divide the results of fruit detection and yield estimation into the subheadings.

10. It is suggest the include a detailed error analysis that quantifies the impact of leaf occlusion, point cloud inaccuracies, and other sources of error on the results.

11. The algorithm has been tested on a small orchard area. It is possible to scale it to larger or denser orchards.

Reviewer 2 Report

Comments and Suggestions for Authors

This study investigates fruit detection and yield mass estimation in apple orchards using UAV-based photogrammetric RGB dense cloud data. The research is timely and relevant, addressing the increasing need for automated orchard management and yield prediction methods. The authors employ UAVs with consumer-grade RGB cameras and use photogrammetry to generate 3D dense point clouds for detecting apple fruits and estimating their yield mass. However, the study is limited by noise and occlusion caused by dense foliage, especially in the lower-altitude datasets. The detection model occasionally underestimates or overestimates apple counts, indicating room for refinement. The introduction could benefit from a clearer contextualization of how the findings compare with other UAV or ground-based fruit detection methods. Figures could be more legible, especially those showing block-level yield estimations. This manuscript can be accepted after the following revisions are made carefully.

1)      Minor grammatical inconsistencies could be refined for better readability.

2)      Introduction: Provide a brief comparison with existing UAV and ground-based detection technologies.

3)      Materials and Methods: Include a summary table of parameter settings for the fruit detection model.

4)      Results: Enhance the clarity and resolution of the figures showing orchard block results.

5)      Discussion: Add a critical reflection on the limitations caused by occlusion and noise in the data. Expand the discussion section to better contrast findings with other UAV and ground-based detection systems. Provide a deeper analysis of error sources, especially the influence of lighting conditions and leaf occlusions.

6)      Conclusion: Suggest emphasizing future research directions, including potential applications of AI/ML techniques. Consider testing advanced machine-learning algorithms (e.g., deep learning) for point cloud classification to improve fruit detection accuracy.

Reviewer 3 Report

Comments and Suggestions for Authors

Overall Assessment

This study presents a method to estimate preharvest conditions in orchards using low-cost drone technology equipped with a consumer-grade RGB sensor.

It is an engaging study with strong potential for practical applications.
The manuscript is written in high-quality English.
Publication is recommended following minor revisions.

Abstract
Clear and comprehensive.

Introduction
Provides a detailed and thorough introduction.
Presents well-defined hypothesis.

Materials and Methods
The methods are clearly described and easy to follow.
Includes a comprehensive flowchart.

•    Figure 1: The study site location is not clearly marked on the upper-right map. It should be highlighted in a distinct and easily visible colour.
•    Table 1: Overlapping numbers and text make it difficult to read; this requires correction.

Results
Results are detailed and well-presented.
•    Figures 4, 5, 7, 8, 9, and 10: The font size is too small and difficult to read. This should be improved.

Discussion
Provides a detailed and thorough discussion.

Conclusions
Presents a concise and well-written conclusion.

Round 2

Reviewer 1 Report

Comments and Suggestions for Authors

Accept in present form